The mitochondrial genome sequences of eleven leafhopper species of Batracomorphus (Hemiptera: Cicadellidae: Iassinae) reveal new gene rearrangements and phylogenetic implications

Lu Jikai 1
Wang Jiajia 2
Dai Renhuai dmolbio@126.com 1
Wang Xianyi 3
1 Institute of Entomology, Guizhou University; The Provincial Key Laboratory for Agricultural Pest Management Mountainous Region , Guiyang , Guizhou , China
2 College of Biology and Food Engineering, Chuzhou University , Chuzhou , Anhui , China
3 Immune Cells and Antibody Engineering Research Center of Guizhou Province, Key Laboratory of Biology and Medical Engineering, School of Biology and Engineering, Guizhou Medical University , Guiyang , Guizhou , China
Zhang Lin
Electronic publication date: 2024 Oct 22
Publication date: 2024
Volume: 12
Electronic Location ID: e18352
Received 2023 Nov 28; Accepted 2024 Sep 27
Copyright: ©2024 Lu et al.
Copyright year: 2024
Copyright holder: Lu et al.
License: This is an open access article distributed under the terms of the Creative Commons Attribution License, which permits unrestricted use, distribution, reproduction and adaptation in any medium and for any purpose provided that it is properly attributed. For attribution, the original author(s), title, publication source (PeerJ) and either DOI or URL of the article must be cited.
License URL: https://creativecommons.org/licenses/by/4.0/

Keywords: Batracomorphini, Iassinae, Leafhopper, Mitogenome, Phylogenetic analysis

Funding: National Natural Science Foundation of China No. 32160119 Program of Excellent Innovation Talents, Guizhou Province, Chin Grant number 20206003-2 This research was supported by the National Natural Science Foundation of China (No. 32160119); and the Program of Excellent Innovation Talents, Guizhou Province, China (Grant number 20206003-2). The funders had no role in study design, data collection and analysis, decision to publish, or preparation of the manuscript.

==============================
Batracomorphus is the most diverse and widely distributed genus of Iassinae. Nevertheless, there has been no systematic analysis of the genome structure and phylogenetic relationships of the genus. To determine the characteristics of the mitogenomes of Batracomorphus species as well as the phylogenetic relationships between them, we sequenced and compared the mitogenomes of 11 representative Batracomorphus species. The results revealed that the mitogenomes of the 11 Batracomorphus species exhibited highly similar gene and nucleotide composition, and codon usage compared with other reported mitogenomes of Iassinae. Of these 11 species, we found that the mitogenomes of four species were rearranged in the region from trnI-trnQ-trnM to trnQ-trnI-trnM, whereas the remaining species presented a typical gene order. The topologies of six phylogenetic trees were in agreement. Eurymelinae consistently formed paraphyletic groups. Ledrinae and Evacanthinae formed sister taxa within the same clade. Similarly, Typhlocybinae and Mileewinae consistently clustered together. All phylogenetic trees supported the monophyly of Iassinae, indicating its evolutionary distinctiveness while also revealing its sister relationship with Coelidiinae. Notably, the nodes for all species of the genus Batracomorphus were well supported and these taxa clustered into a large branch that indicated monophyly. Within this large branch, four Batracomorphus species with a gene rearrangement (trnQ-trnI-trnM) exhibited distinctive clustering, which divided the large branch into three minor branches. These findings expand our understanding of the taxonomy, evolution, genetics, and systematics of the genus Batracomorphus and broader Iassinae groups.

Introduction

Cicadellida, the most diverse family of Hemiptera, comprises many species that are capable of damaging cash crops. To date, there are more than 23,000 described species in this family that are distributed into 25 subfamilies (Krishnankutty et al., 2016; Wang et al., 2020a; Xue, Dietrich & Zhang, 2020). Of these subfamilies, Iassina consists of 12 tribes and 155 genera and covers >2,000 known extant species (Dietrich, Nguyen & Pham, 2020; Domahovski & Cavichioli, 2019; Domahovski, 2019; Domahovski et al., 2019; Domahovski et al., 2020; Domahovski & Cavichioli, 2022; Goncalves, Dietrich & Takiya, 2020; Krishnankutty et al., 2016; Laranjeira et al., 2022; Wang et al., 2020b; Yang & Dai, 2022), with members widely distributed throughout zoogeographic regions (Hu et al., 2021; Wang et al., 2020a). Like the other leafhoppers, Iassines primarily reside in the arbor, whereas some leafhoppers inhabit shrubs, herbaceous plants, and grasses. Recent studies have shown that Batracomorphus Lewis (1834) is one of the most diverse genera of the subfamily Iassinae, with >360 described extant species, of which 38 have been recorded in China as pests of economically important crops, including fruit trees (e.g., grapes, apples, and pears) and crops (e.g., rice, tomato, and potato) (Grylls, 1979; Hu et al., 2021; Jian & Kaiyu, 2010; Knight, 1983; Krishnankutty et al., 2016; Wang et al., 2010). The morphological characteristics of this genus are extremely similar across species, which makes accurate identification challenging, and often necessitating dissection of the male genitalia. Consequently, it is important to identify species and their taxonomic relationships within this genus for effective pest management. However, the vast majority of systematic studies on this genus have been conducted through traditional taxonomy, which lacks molecular data for result verification. Although several systematic studies based on molecular DNA data were published recently, only a few species of Batracomorphus have been included (Dietrich et al., 2001; Hu et al., 2021; Krishnankutty et al., 2016; Wang et al., 2020a; Wang et al., 2020b). Therefore, a more extensive dataset that covers a broad range of Batracomorphus species is warranted to examine the phylogenetic relationships between the members of this genus.

With the recent development of high-throughput sequencing technology, mitochondrial genomes (mitogenomes) have become routine molecular markers. The number of mitogenomes in the family Cicadellidae is growing annually, and these data are increasingly utilized in phylogenetic studies (Jiang et al., 2022; Wang et al., 2020a; Wang et al., 2022; Yan et al., 2022; Yuan et al., 2021). Moreover, gene rearrangement has become a hot topic in mitogenomic studies, providing information on phylogenetic inference. However, studies are still minimal on the mitogenome characteristics of Batracomorphus. Until now, only two species have been deposited in the NCBI database (Wang et al., 2020b), which account for 0.5% of all species in this genus; thus the structural characteristics of the mitogenomes of Batracomorphus are not yet fully understood.

To elucidate the structural features of the mitogenomes of Batracomorphus leafhoppers and the phylogenetic relationships between them, we sequenced and compared the mitogenomes of 11 representative Batracomorphus species in China. Additionally, we constructed phylogenetic trees based on the different associations of mitochondrial genes to determine the taxonomic position of this group and its phylogenetic relationship with other leafhopper families. This provides additional data to support the rationale for the existing categorizing system.

Materials and Methods

Insect sampling and DNA extraction

Adult male leafhopper samples were collected from the field using a sweep net during the daytime or via light-trapping at night. The samples were preserved directly in absolute ethanol and transported to the laboratory, where preliminary identification was performed based on morphological characteristics. Each species sample was placed in a 2-ml centrifuge tube containing absolute ethanol and stored in a −20 °C freezer until identification and DNA extraction. Eleven Batracomorphus species were first identified morphologically based on the male genitalia (Knight, 1983; Hu et al., 2021). Total DNA was extracted from the muscle tissue of adult males using QIAamp® DNA Micro Kit (50) according to the manufacturer’s instructions. The extracted genomic DNA was stored at −80 °C and used for sequencing. Voucher specimens with the male genitalia and DNA samples were deposited at the Institute of Entomology, Guizhou University, Guiyang, Guizhou, China.

Sequencing, assembly, and bioinformatics analyses

The genomic DNA of the 11 species were sequenced using the Illumina NovaSeq 6000 platform (Berry Genomic, Beijing, China) with a 150-bp paired-end setting, yielding 6 GB of raw data per sample. The resulting clean data were quality-trimmed and assembled using the “Mapping to Reference” function of Geneious Prime v.11.0.4, with the mitogenome sequence of B. lateprocessus (GenBank: MG813489) as a reference. The assembled mitogenome sequences were compared with the homologous sequence of T. arisana (GenBank: NC036480), which was retrieved from GenBank and subjected to a BLAST search on the NCBI to confirm the accuracy of the sequence (Kearse et al., 2012). We used the MITOS web server (http://mitos.bioinf.uni-leipzig.de/index.py) to preliminarily annotate the assembled mitogenome sequences based on the genetic code for invertebrates (Bernt et al., 2013). The positions of 22 tRNA genes were reconfirmed and predicted via ARWEN v.1.2 and tRNAscan-SE v.1.21 (Laslett & Canbäck, 2008; Schattner, Brooks & Lowe, 2005). The rRNA genes (16S rRNA and 12S rRNA genes) were determined based on the locations of the adjacent tRNA genes (trnV and trnL2) and through a comparison with other Cicadellidae mitogenome sequences in NCBI. The start and stop codons of the protein-coding genes (PCGs) were determined by the ORF Finder using invertebrate mitogenomes listed on the NCBI website (https://www.ncbi.nlm.nih.gov/orffinder/). The base composition of the PCGs and relative synonymous codon usage (RSCU) were analyzed using MEGA v.11 (Tamura, Stecher & Kumar, 2021). Strand asymmetry was calculated using the following formulas: AT bias = (A − T)/(A + T) and GC bias = (G − C)/(G + C) (Perna & Kocher, 1995).

Sequence extraction, alignment, and concatenation

Phylogenetic analysis was performed using the mitogenomes of 11 Batracomorphus species examined in this study and 102 species downloaded from NCBI, with 6 species each of Cicadellidae and Cercopidae as outgroups (Table S1). PCGs and rRNA genes were extracted from the mitogenomes of 113 species using ‘Extract Annotations’ in Geneious Prime v.11.0.4. Each PCG was compared based on the G-INS-I (accurate) strategy and codon alignment model using the MAFFT component in PhyloSuite software (Katoh, Rozewicki & Yamada, 2019). The 16S and 12S rRNA genes were aligned using the Q-INS-I strategy in MAFFT v.7 (https://mafft.cbrc.jp/alignment/server/). The aligned sequences were trimmed using ‘trimAI’ with PhyloSuite software. The trimmed sequences were combined to form different datasets using MEGA v.11 (Tamura, Stecher & Kumar, 2021). Each dataset included the following information: (1) a PCG12 matrix, including 13 PCGs with 7,240 nucleotides at the first and second codon positions (PCG12); (2) a PCG12 rRNA matrix comprising 13 PCGs first and second codon positions and two rRNA genes (16S and 12S rRNA genes), which included 8,953 nucleotides; and (3) a PCGrRNA matrix containing all codon positions of 13 PCGs and two rRNA genes (16S and 12S rRNA genes), which included 12,573 nucleotides.

Phylogenetic analysis

To increase the reliability of each phylogenetic tree, the optimal model for each data set was calculated using PartitionFinder v.2.1.1 to obtain the optimal model for each gene segment (Miller, Pfeiffer & Schwartz, 2010). The three datasets were converted into ‘nex’ and ‘phy’ formats using Mesquite v.2.75 software, respectively. Phylogenetic analyses were performed using IQ-tree v.1.6.12 (Nguyen et al., 2015) and MrBayes v.3.2.7 (Huelsenbeck & Ronquist, 2001). A maximum likelihood (ML) phylogenetic tree was constructed using IQ-TREE with 10,000 iterations via ultra-fast bootstrap approximation. A Bayesian inference (BI) phylogenetic tree was generated using MrBayes v.3.2.7. BI analysis was performed using the default settings to simulate four independent runs for an aggregate of 100 million generations with a sample taken once per 1,000 times. When the convergence value reached <0.01, the operation was stopped. The tree was stored every thousand generations, and the first 25% were discarded according to burning parameters to form a 50% merge tree. The phylogenic trees were imaged using Figtree v.1.4.2.

Results

Organization of the mitochondrial genomes

In this study, mitogenome data were obtained for 11 Batracomorphus species. These mitogenomes varied in length between 14,870 bp in B. chlorophane to 15,385 bp in B. rinkihonis (Fig. 1; Fig. S1). The mitogenome structure of the genus was stable and composed of double-stranded closed-loop DNA molecules, including 37 typical genes and an A + T-rich region. The nucleotide composition of the mitogenomes of the 11 species was as follows: A = 46.3–48%, T = 32.2–34.2%, G = 7.8–8.6%, and C = 11.2–12.1%, respectively (Table S2). The nucleotide composition of the mitochondrial genes of Batracomorphus species was significantly biased toward adenine (A) and thymine (T). Of these, the A + T content of B. lineatus was the lowest, accounting for 79.3% of the entire mitogenome, whereas the A + T content of B. nigromarginattus accounted for up to 81% of the mitogenome. All species showed a positive AT bias (0.15 to 0.19) and a negative GC bias (−0.86 to −0.16) (Fig. 2; Table S2), which are common features of Cicadellidae mitogenomes (Wang et al., 2018; Wang et al., 2021; Wang, Wang & Dai, 2021a; Wang, Wang & Dai, 2021b).

Figure 1 Circular maps of the mitochondrial genomes of 11 species in Batracomorphus.

Figure 2 Three-dimensional scatter plot of AT-skew, GC-skew, and A + T content of the mitochondrial genes in 11 species of Batracomorphus.

The uppercase letters represent the species’ names, and different colors represent different genes.

PCGs and codon usage

This study showed that the tandem lengths of 13 PCGs from the 11 examined species ranged from 10,898 to 10,919 bp. The tandem lengths of 13 PCGs from the four species with gene rearrangements were 10,919 bp (B. nigromarginattus and B. allionii; encoding 3,628 amino acids excluding the terminator), 10,918 bp (B. lineatus; encoding 3,627 amino acids, excluding the terminator), and 10,916 bp (B. extentus; encoding 3,627 amino acids, excluding the terminator). In the 11 mitogenomes, atp8 was the shortest PCG (150 or 153 bp), whereas nad5 was the longest PCG (1,680, 1,677, or 1,678 bp). Among the 13 PCGs of the 11 Batracomorphus mitogenomes, nine PCGs were encoded on the J-chain, and four PCGs were encoded on the N-chain (Tables S4–S14). The A + T content of the 13 PCGs was between 77.9% (B. lineatus) and 80.1% (B. nigromarginattus), which showed a clear AT preference (AT bias ranged from −0.13 (B. curvatus) to −0.09 (B. allionii), with an average of −0.112) and a slight GC bias (GC bias ranged from −0.04 (B. extentus) to 0.06 (B. allionii), with an average of −0.037) (Fig. 2; Table S2).

Previous studies have shown that the vast majority of PCGs in Cicadellidae use ATN (N represents A, T, C, or G) as the start codon, of which ATG is the most frequently used (Du et al., 2021; Lin, Huang & Zhang, 2021; Wang et al., 2020a). The current study results also demonstrated that ATG was the most frequently used start codon, with the start codons for cox1, cox3, nad4L, and cytb being ATG across all 11 species. Notably, the start codon of atp8 in this genus was TTG. The stop codons of the PCGs of Cicadellidae are typically TAA or TAG, and sometimes the incomplete codon T--. The stop codons of nad1, nad2, cox1, and atp8 in this study were TAA, the stop codon of cox2 was the incomplete codon T--, and the stop codons of the other PCGs were TAA, TAG, or T-- (Tables S4–S14). We calculated and mapped the codon usage number and relative synonymous codon usage (RSCU) of the PCGs for the 11 species (Table S3). The most used codons included AUA (361 to 410), UUU (354 to 391), AUU (330 to 379), and UUA (285 to 346). The most used amino acids were Phe (F), Leu (L), Ile (I), Met (M), and Ser (S) (Table S3), consistent with other studies (Wang et al., 2019; Zhou, Dietrich & Huang, 2020).

tRNAs and rRNAs

We analyzed 22 tRNA genes in the mitogenomes of the 11 species and found that the tRNA genes spanned a region extending between 1,403 bp (B. allionii) and 1,422 bp (B. matsumurai). The shortest single tRNA gene of all species was trnG (53 to 60 bp), except for B. nigromarginattus, in which the shortest tRNA gene was trnC (60 bp). The longest tRNA gene was trnK (71 to 72 bp), except for B. extentus, in which the longest tRNA gene was trnM (72 bp). Consistent with the majority of leafhopper mitogenomes, the tRNAs of Batracomorphus were scattered throughout the mitochondria: the J-chain encoded 14 tRNAs, whereas the remaining tRNAs were encoded by the N-chain (Tables S4–S14). In Batracomorphus species, the A + T content of all tRNAs was between 78.7% (B. allionii) and 80.5% (B. rinkihonis), with a slight AT bias and a pronounced GC bias (Fig. 2; Table S2).

The length of the rRNA gene regions (16S and 12S rRNA genes) ranged from 1,872 bp (B. lineatus, B. matsumurai, and B. notatus) to 1,910 bp (B. rinkihonis). The positions of all rRNA genes in the mitogenomes of the species were determined by aligning with neighboring species (B. lateprocessus). The lengths of all 16S rRNA genes ranged from 1,165 bp (B. extentus and B. lineatus) to 1,177 bp (B. cornutus), and they were located between trnL1 and trnV. All 12S rRNAs genes were located between trnV and the control region, and the length of these genes was between 700 bp (B. fuscomaculatus) and 738 bp (B. rinkihonis) based on an alignment with neighboring species (Tables S4–S14). The A + T content of the rRNAs ranged from 82.5% (B. chlorophane) to 83.4% (B. nigromarginattus, B. rinkihonis). AT and GC biases were evident. The AT bias values were between −0.23 (B. chlorophane) and −0.18 (B. allionii), whereas the GC bias values ranged between 0.22 (B. curvatus) and 0.26 (B. fucomaculatus) (Fig. 2; Table S2).

A + T-rich regions

Studies have revealed that the control region, an indispensable noncoding fragment, is the beginning region of replication and initiates the transcription and replication of mitogenomes (Zhou, Dietrich & Huang, 2020). The placement of the control region within Batracomorphus species was generally constant, typically positioned after trnI and before the 12S rRNA gene. Compared with other mitochondrial genes of Batracomorphus, the control region was the least conserved, with no clear regularity reported in its length, which was the crucial reason for the discrepancy in mitochondrial length (Fig. S1). The length of the control regions ranged from 698 bp (B. chlorophane) to 1,212 bp (B. curvatus). The A + T content in the control regions of the 11 species was between 83% (B. curvatus) and 88.6% (B. cornutus). The large variation in the A + T content was mainly due to the length of the A + T-rich region (Fig. 2). A clear TA preference (AT bias ranged from 0.07 (B. rinkihonis) to 0.12 (B. chlorophana)) and a slight CG bias (CG bias ranged from −0.29 (B. nigromarginattus) to 0.15 (B. notatus)) were observed (Fig. 2; Table S2). The size and number of repetitive sequences in the control region of the mitogenomes of Batracomorphus were variable. There were one to three repeats in each Batracomorphus species, with each fragment repeating two to three times; the length of the repeats ranged from 42 bp (B. nigromarginattus) to 203 bp (B. notatus) (Fig. 3). The repetitive sequences in the control region of the mitochondrial gene did not show any regularity, which is consistent with the control region of other insects (Jiang et al., 2022; Wang et al., 2021; Wang, Wang & Dai, 2021a; Wang, Wang & Dai, 2021b).

Figure 3 The organization of the structure of the control area of 11 species in Batracomorphus.

Gene rearrangements and intergenic spacers

The arrangement of mitochondrial genes is relatively conserved in insects; however, some groups have rearrangements compared with the Drosophila mitochondrial gene sequence as a model (Liu et al., 2022; Ye, Li & Xie, 2021; Zhang et al., 2021). The gene arrangement of the mitogenomes of Cicadellidae was highly conserved. No rearrangement of the 13 PCGs and 2 rRNAs was observed, whereas only a few of the 22 tRNAs were rearranged, with a small probability in a few species (Du et al., 2017; Du, Dietrich & Dai, 2019; Mao, Yang & Bennett, 2017; Song, Zhang & Zhao, 2019; Wang et al., 2020b). In this study, the tRNA gene rearrangement events of four species (B. allionii, B. extentus, B. lineatus, and B. nigromarginatus) may have occurred in three steps. First,the gene duplication produced a cluster of duplicated genes (trnI-trnQ-trnM-trnI-trnQ-trnM). Subsequently, the upstream duplicated trnI gene and downstream duplicated trnQ gene were randomly lost. Simultaneously, the random loss of the upstream duplicated trnM gene generated a 43 to 47 bp noncoding region. Finally, a new gene order trnQ-trnI-trnM was generated (Fig. 4).

Figure 4 The hypothetical process of trnI-trnQ-trnM translocation in the duplication-random loss (TDRL) model.

An “×” indicates partial random loss of the replicated gene; NC, genes outside the rearrangement region.

tRNA gene rearrangements have also been reported in the mitochondrial genes of several other subfamilies in the leafhopper family. To further understand the evolution of mitochondrial gene rearrangements in the leafhopper subfamily, we mapped the five tRNA rearrangement models of the leafhopper family into a putative phylogenetic tree (Fig. 5). Three types of rearrangements of tRNA gene clusters were found in the Deltocephalinae subfamily. In MR1, the tRNA gene cluster trnW-trnC-trnY rearranges to trnC-trnW-trnY (Cicadulina mbila, Macrosteles quadrilineatus, and M. quadrimaculatus) (Du, Dietrich & Dai, 2019; Mao, Yang & Bennett, 2017; Song, Zhang & Zhao, 2019). In MR2, the tRNA gene cluster trnW-trnC-trnY rearranges to trnY-trnW-trnC in Japananus hyalinus (Du et al., 2017). In MR3, the tRNA gene cluster trnA-trnR-trnN-trnS1-trnE-trnF rearranges to trnR-trnE-trnF-trnA-trnN-trnS1 in Stirellus bicolor, Elymana sp., and Osbornellus sp. (Song, Zhang & Zhao, 2019). In MR4, the tRNA gene cluster trnI-trnQ-trnM rearranges to trnQ-trnI-trnM as found in Cicadellinae (Cofana unimaculata) and Iassinae (T. arisana) (Song, Zhang & Zhao, 2019; Wang et al., 2020b). Additionally, trnI gene duplication occurs in the subfamily Eurymelinae (Liocratus salicis) (Wang et al., 2018), which is a duplication of trnI-trnQ-trnM into trnI-trnI-trnQ-trnM (MR5). In this study, the mitogenomes of Batracomorphus were rearranged in the MR4 model, which sheds light on the mitochondrial gene rearrangements in the Iassinae and leafhopper families.

Figure 5 The gene rearrangements and gene spacers are identified in the mitogenomes of Cicadellidae.

Different colors represent different genes. Dashed boxes of different colors indicate different rearrangement models.

The gene rearrangement in the mitogenomes of Cicadellidae was relatively compact, with fewer and shorter overlapping and spacer regions between the genes, but a few species with longer intergenic regions were also observed after gene rearrangements (B. lateprocessu, Trocnadella arisana, and Krisna rufimarginata). Through a comparative analysis of the mitogenomes of the 11 species sequenced in this study, we found varying degrees of gene overlap and spacing. Except for the four species with gene rearrangements, the number of overlapping regions was 14–20 in all species, whereas the length of spacers was <25 bp. Due to the gene rearrangement from trnI-trnQ-trnM to trnQ-trnI-trnM in the four species of Batracomorphus genus resulted in a relatively long intergenic sequence between trnQ and trnI genes, with an interval length of 43 to 47 bp (Fig. S2).

Phylogenetic analyses

In the present study, we examined the phylogenetic relationships among species Cicadellidae based on the sequenced mitogenomes of 11 Batracomorphus species and 102 other species available in the NCBI database (Table S1). The results of six phylogenetic trees with well-supported bootstrap replications were obtained using three datasets (PCG12, PCG12rRNA, and PCGrRNA) and two analysis methods (Bayesian and ML) (Fig. 6; Figs. S3–S7). The results indicated that the phylogenetic relationship based on the PCG12 data was more stable than that based on the other two datasets. The reason for this difference may be because of the saturation effect at the 3rd codon position in the PCG123 dataset in the mitogenome phylogeny. However, the phylogenetic relationship between the subfamilies was unstable and the topological structure within some of the subfamilies differed. All phylogenetic trees revealed that Eurymelinae and Hylicinae form a paraphyletic group. Macropsini and Idiocerini did not form a sister group relationship. Macropsini was stably clustered with Hylicinae, whereas Idiocerini was clustered with Treehopper and Megophthalminae (BS > 91, Bayesian PP = 1). Idiocerini also showed an apparent paraphyletic group, and Idioscopus species belonging to this family could not be clustered into a monophyletic group. In addition, the current study results indicate that Ledrinae is a sister group of Evacanthinae. Typhlocybinae and Mileewinae (BS > 67, Bayesian PP > 0.72) formed a branch showing a sister group relationship. Iassinae and Coelidiinae (BS = 1, Bayesian PP = 1) maintained a stable sister relationship; and Deltocephalinae formed an independent branch at the root of the tree, which is essentially consistent with the finding of previous studies (Cao et al., 2022; Dietrich et al., 2017; Hu et al., 2021; Skinner et al., 2020).

Figure 6 Phylogenetic tree of Cicadellidae inferred by Bayesian analyses of the first and second codon positions of the 13 protein-coding genes.

Based on the analyses, the genera involved in belonging to the subfamily Iassinae were monophyletic groups, and most nodes were strongly supported (BS > 91, Bayesian PP > 0.93) (Fig. 6; Figs. S3–S7). Of these, Krisna and Gessius were clustered into a clade and formed a sister group, whereas Trocnadella and Batracomorphus formed a sister group within the same clade. The species involved in Batracomorphus formed a large branch, whereas four species (B. lineatus, B. nigromarginattus, B. allionii, and B. extentus) with a rearrangement (trnI-trnQ-trnM rearranged to trnQ-trnI-trnM) were clustered into a small branch in the middle, which divided Batracomorphus into three distinct clades. Phylogenetic analyses of Batracomorphus species strongly supported the finding that B. alllionii and B. nigromarginattus were closely related as a sister group, and formed a closely related species group alongside B. extentus and B. lineatus. Meanwhile, B. curvatus emerged as a sister group with B. lateprocessus and B.matsumurai in all analyses. Unfortunately, the status of B. fuscomaculatus and B. cornutus in the phylogenetic relationships remains unclear. Nevertheless, additional specimens and molecular data are warranted to further elucidate the phylogeny of the genus.

Discussion

Mitochondrial genome structure

Our study provides the first comprehensive analysis of the mitogenomes of Batracomorphus species, which may greatly expand the phylogenetic understanding of the genus Batracomorphus. We conducted a comparative analysis of the characteristics of Batracomorphus mitogenomes, such as genome length, gene arrangement, nucleotide composition, relative codon usage of PCGs, and noncoding regions. The variation in mitochondrial gene size among the 11 species of Batracomorphus was relatively small and was primarily determined by the length of the control region (Fig. S1), which is also typical in most other insect species (Li et al., 2022; Lu et al., 2023). The mitochondrial genes of this genus exhibited a notable AT skew in nucleotide composition. This skew may arise due to selection pressures or asymmetries in base mutation during replication and transcription processes (Bogenhagen & Clayton, 2003; Brown et al., 2005). Previous studies have shown that the 13 PCGs of the leafhopper family primarily use the standard triplet codon ATN, whereas ATG was the most frequently used start codon in Batracomorphus. The most frequently used stop codon was TAA. However, the cox2 and nad5 genes of Batracomorphus contained an incomplete stop codon T--, which is particularly common in insect cox2 and nad5 genes and thus requires post-transcriptional modification during mRNA maturation (Ge et al., 2023; Zhang et al., 2020a; Zhang et al., 2020b).

Gene rearrangements in the mitogenomes are extremely rare in the leafhopper family. Currently, only five model tRNA gene rearrangements have been reported in the mitogenomes of several other leafhopper subfamilies (Fig. 5). To date, the molecular mechanisms used to explain mitochondrial gene rearrangements in insects mainly include duplication-random loss (TDRL), duplication-nonrandom loss, and recombination (Ge et al., 2023; Wang, Bai & Dong, 2022; Wang et al., 2021). The TDRL model assumes that a new sequence of mitochondrial genes arises from the duplication of mitochondrial genes, followed by a loss of their repetitive sequences, which has been favored by many researchers (Macey et al., 1997; Moritz & Brown, 1987). The rearrangement of the region from trnI-trnQ-trnM to trnQ-trnI-trnM, which was observed in four species of Batracomorphus in the present study, resulted from the rearrangement mechanism of the TDRL model, which provides further evidence regarding the novel gene rearrangement explained by the TDRL model. We hypothesized that the trnI-trnQ-trnM genes of the involved species undergo random loss after duplication, and a 43 to 47 bp noncoding region is generated between trnQ and trnI genes during the loss process. With increased analysis of mitogenomes and expanded sampling of taxonomic units, new rearrangements may be discovered that will not only expand our understanding of the evolution of mitogenomes in the leafhopper family but will also be helpful to elucidate the evolutionary relationships between different taxonomic groups.

Implications for the phylogeny

The phylogenetic analysis results revealed a well-supported monophyletic status among the subfamilies in Cicadellidae, except for Eurymelinae, which exists as a paraphyletic group. Macropsini and Idiocerini exist as a cluster together to present a sister group relationship, which was different from the results of a previous study (Xue, Dietrich & Zhang, 2020). Macropsini consistently clustered with Hylicinae, whereas Idiocerini clustered with Treehopper and Megophthalminae to form a close phylogenetic relationship. Therefore, this study supports that Treehoppers originated from paraphyletic Cicadellidae, whereas more species data are required to confirm the placement of Idiocerini and Macropsini in Eurymelinae (Wang et al., 2018; Xue, Dietrich & Zhang, 2020; Wang, Wang & Dai, 2021a; Wang et al., 2022). The monophyly of Iassinae is controversial due to the unstable relationship of Gyponinae, Bythoniinae, and Iassinae. Linnavuori & Quartau (1975) combined the three subfamilies into Iassinae based on morphological characteristics. Dietrich et al. (2001) constructed a phylogenetic tree of Membracoidea based on the 28S rRNA gene sequence, demonstrating that Gyponinae forms a sister group with Iassinae; however, Bythoniinae was not placed in the same clade with these two subfamilies. Krishnankutty et al. (2016) analyzed the phylogenetic relationships of Iassinae based on morphological characteristics and molecular data and found that Iassinae forms a monophyletic group, which suggests that Gyponinae was derived from Iassinae. Hu et al. (2021) analyzed transcriptome data, which revealed that Iassinae was consistently polyphyletic. We inferred the phylogenetic relationships of Iassinae using various strategies, and demonstrated that Iassinae and Coelidiinae maintain a stable sister relationship, which is also supported by other studies (Wang et al., 2019; Du et al., 2021). The results of our phylogenetic analyses revealed that Iassinae forms a monophyletic group, which is consistent with previous studies that are based on morphological and molecular data (Krishnankutty et al., 2016; Wang et al., 2020b; Wang, Wang & Dai, 2021b), despite inconsistency with a previous transcriptome study (Hu et al., 2021). The topology among the tribes of Iassinae was identical to the phylogenetic relationship previously inferred based on morphological and molecular data (Krishnankutty et al., 2016; Wang et al., 2020b). Compared with previous studies, our analysis provides more comprehensive information on the evolution and phylogenetic relationships of Batracomorphus species. Notably, Batracomorphus is a monophyletic group, and the topological structure could be divided into three distinct clades. Meanwhile, the results of this study supported Krishnankutty’s view that established Batracomorphini based on morphological characteristics and molecular fragments using Batracomorphus as the model genus (Krishnankutty et al., 2016). Our findings elucidated the evolutionary relationship among the 11 species of Batracomorphus in China. These phylogenetic analyses indicate that B. chlorophana and B. notatus, B. nigromarginattus and B. extentus, and B. lateprocessus and B. matsumurai form a sister group relationship. Nonetheless, the number of species used in this study was inadequate to represent the whole genus. Therefore, more species need to be added to further clarify the evolutionary relationship between the Batracomorphus species.

Conclusions

A total of 11 Batracomorphus mitogenomes were sequenced and analyzed in the present study, and a gene rearrangement was identified in four species (B. allionii, B. extentus, B. lineatus, and B. nigromarginatus). tRNA gene rearrangement events may include random gene loss after trnI-trnQ-trnM replication to form trnQ-trnI-trnM based on the TDRL model. Furthermore, we examined the phylogeny of Iassinae based on mitogenomes. Our results support that Iassinae is a monophyletic group, demonstrating a sister group relationship between Iassinae and Coelidiinae. We conclude that Batracomorphus is a monophyletic group; however, four species were arranged into branches, and the genus was divided into three distinct clades. The results suggest that B. chlorophana and B. notatus, B. nigromarginattus and B. extentus, and B. lateprocessus and B. matsumurai formed a sister group relationship. The results of this paper provide a basis for a more detailed exploration of the phylogenetic relationships of Iassinae in future studies.

Supplemental Information

Figure S1 Mitochondrial genome length of 11 species of Batracomorphus

Figure S2 Intergenic spaces associated with gene rearrangement in the mitochondrial genomes of 4 Batracomorphus species

Figure S3 Phylogenetic tree of Cicadellidae inferred by Bayesian analyses of the 1st and 2nd codon locations of 13 PCGs and 2 rRNA genes

Figure S4 Phylogenetic tree of Cicadellidae inferred by Bayesian analyses of 13 PCGs and 2 rRNA genes

Figure S5 Phylogenetic tree of Cicadellidae constructed using ML method according to the 1st and 2nd codon locations of 13 PCGs

Figure S6 Phylogenetic tree of Cicadellidae constructed using ML method according to the 1st and 2nd codon locations of 13 PCGs and 2 rRNA genes

Figure S7 Phylogenetic tree of Cicadellidae constructed using ML method according to of 13 PCGs and 2 rRNA genes

Table S1 Sources and information used for the polygenomic analysis

Table S2 Base content and skew of various types of mitochondrial genes in the Batracomorphus.

Table S3 The relative synonymous codon usage of the PCGs of 11 Batracomorphus mitochondrial genomes

Table S4–S14 Mitochondrial genome organization of Batracomorphus

Table S15 Collection information of specimen in the present study

Note: B. rinkihonis indicates Batracomorphus rinkihonis, B. notatus indicates Batracomorphus. B. notatus , B. nigromarginattus indicates Batracomorphus nigromarginattus, B. matsumurai indicates Batracomorphus matsumurai, B. lineatus indicates Batracomorphus lineatus, B. fuscomaculatus indicates Batracomorphus fuscomaculatus, B. extentus indicates Batracomorphus extentus, B. curvatus indicates Batracomorphus curvatus, B. cornutus indicates Batracomorphus cornutus, B. chlorophana indicates Batracomorphus chlorophana, B. allionii indicates Batracomorphus allionii.

We sincerely thank Likun Zhong, Bin Yan, Ling Qu, Chao Zhang, and Qin Zuo for collecting the specimens for this study. We also thank Jiapeng Yang for his guidance on drawing techniques. We thank Meishu Guo, Die Liu, Yanqiong Yang, and Min Li for the English language modifications in the manuscript.

Additional Information and Declarations

Competing Interests

Author Contributions

DNA Deposition

Data Availability

The authors declare they have no competing interests.

Jikai Lu conceived and designed the experiments, performed the experiments, analyzed the data, prepared figures and/or tables, authored or reviewed drafts of the article, and approved the final draft.

Jiajia Wang conceived and designed the experiments, performed the experiments, analyzed the data, authored or reviewed drafts of the article, and approved the final draft.

Renhuai Dai conceived and designed the experiments, authored or reviewed drafts of the article, and approved the final draft.

Xianyi Wang conceived and designed the experiments, authored or reviewed drafts of the article, and approved the final draft.

The following information was supplied regarding the deposition of DNA sequences:

Batracomorphus fuscomaculatus: OQ873422, Batracomorphus notatus: OQ873426, Batracomorphus chlorophana: OQ873418, Batracomorphus cornutus: OQ873419, Batracomorphus rinkihonis: OQ873427, Batracomorphus allionii: OQ873417, Batracomorphus nigromarginattus: OQ873425, Batracomorphus lineatus: OQ873423, Batracomorphus extentus: OQ873421, Batracomorphus matsumurai: OQ873424, Batracomorphus curvatus: OQ873420.

The following information was supplied regarding data availability:

The data are available in the Supplemental Files.

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
