# Peer review of "The mitochondrial genome sequences of eleven leafhopper species of Batracomorphus (Hemiptera: Cicadellidae: Iassinae) reveal new gene rearrangements and phylogenetic implications"

_PeerJ, doi:10.7717/peerj.18352_

## Round 0.1 · original submission · Major Revisions

Thanks for submitting your work to PeerJ.

Please address reviewers' comments to change your manuscript and submit it.

Reviewer 1 ·

Basic reporting

This article presented eleven mitochondrial genomes of Batracomorphus species, reconstructed the phylogeny of Cicadellidae and systemically analyzed their mitogenomic features using mitochondrial genomic data. Meanwhile, they found that the gene sequences of four species in this genus were rearranged from trnI-trnQ-trnM to trnQ-trnI-trnM. This study had provided evidence for the phylogenic analysis within Cicadellidae, as well as improved the understanding of mitogenomic evolution and phylogenetic relationships in Cicadellidae. Although this manuscript is generally well-written, some adjustments must be addressed before it is considered for acceptance.

Experimental design

no comment

Validity of the findings

no comment

Additional comments

[1]
Line 78 yielding 6 Gb of clean data per sample, Clean data or raw data?
It would be informative to add a table to the supplementary documents with information on the cleaning for the illumina reads. The following info could be added to the table.
1. how many reads were there when they were cleaned
2. how was each cleaned?
3. what was the coverage of assembled mitogenomes like?


[2]
The discussion needs to go deeper


[3]
There are many mistakes and unscientific places in the language of the article, which needs to be carefully revised.
Minor details:

Mitochondrial genome, complete mitochondrial genome and mitochondrial genes represent different meanings, need to find the full text and scientific use.

Line 18-19
The six phylogenetic trees of the topology structure The order should be reversed

Line 28
Cicadellidae is the largest and one of the most economically significant families of Hemiptera……. This statement is not clear.

Line 38
Iassines genus ?

Line 50
Cost?

Line 51
the data are ?

Line58-60
This statement is not clear.

Line 94-96
This statement is not clear.

Line 120
mitochondrial gene data

Reviewer 2 ·

Basic reporting

There are still some issues of references citation, such as the Wang et al. 2020b (Wang et al. 2020a is not found before Wang et al. 2020b), etc. Also, when the references were published by the same author in different year, early year should be cited before the latter one. Please check the citation of references in the whole text carefully.

Line 54: “are a” is a grammatical error, please correct.

Line 138: The genus (B. mitogenomes,) cannot be abbreviated here, please correct. And delete the “the” before nad5.

Line 144: Figure 1, it is hard to read the column of respond amino acids, please use another clear one to instead.

Line 153-154: “The most used codons were AUA (361 to 410), UUU (354 to 391), AUU (330 to 379), and UUA 154 (285 to 346). The most used amino acids were Phe (F), Leu (L), Ile (I), Met (M), and Ser (S) (Fig. 2), consistent with 155 other studies (Wang et al. 2019; Zhou et al. 2020).” The result in this sentence is not consistent with the results that Figure 2 shown, please check carefully and correct.

Line 189: correct “There were 1 to 3 repeats in this genus……”, to “There were 1 to 3 repeats in each Batracomorphus species.”

Line 405: This reference is not cited correctly, including the authors, issue, volume, and page, please download and read the reference, and correct its citation.

It seems that the Discussion and Conclusions contain the same content, please revise.

Experimental design

no comment

Validity of the findings

no comment

Reviewer 3 ·

Basic reporting

Overall, this is a substantial and meaningful article. However, I have some questions that need the author's clarification and correction.

Experimental design

1.How specifically was the accuracy of the morphological identification determined based on the Blast results of cox1 in NCBI in lines 71 to 72?

2.What is the body length of Batracommorphus? How many individuals were used to extract DNA from each species?

3.Is the "Mapping to Reference"step using the original sequence R1 or R2, or a combination of R1 and R2?

4.Why use species T. arisana from other genera in lines 80 to 82 to determine the accuracy of the sequence?

5.How did you determine that the rearrangements of the four species in the article are as depicted in Figure 5? Have you verified the rearrangements, for instance, through PCR amplification?

Validity of the findings

1.In line 145, the word "Cicadrllidae" should be "Cicadellidae".

2.Please provide full name when first abbreviating. For example,"TDRL" in line 207 and "MR1" in line 212.

3.In lines 230 to 231, why did the authors describe the intergenic spacers as being between 5 and 12, and then state that it's " below 25 bp"?

4.The description in lines 256 to 257,"B. lineatus and B. alllionii formed a sister group relationship, and formed a closely related species group with B. nigromarginattus and B. extentus," does not align with the results depicted in Figure 7.In addition,the resolution of Figure 7 is too low to clearly discern the species' names upon zooming in.

5.Conclusions typically do not contain references; they should be placed in the appropriate section of the discussion.

---

## Round 0.2 · Major Revisions

Thanks for your work to PeerJ.

Sorry, your work requires further revision. We re-reviewed it, please address these changes and resubmit.

Lin

**Language Note:** The review process has identified that the English language must be improved. PeerJ can provide language editing services - please contact us at [email protected] for pricing (be sure to provide your manuscript number and title). Alternatively, you should make your own arrangements to improve the language quality and provide details in your response letter. – PeerJ Staff

Reviewer 1 ·

Basic reporting

The revised paper has met the publication requirements

Experimental design

no comment

Validity of the findings

no comment

Additional comments

The revised paper has met the publication requirements

Reviewer 2 ·

Basic reporting

no comment

Experimental design

no comment

Validity of the findings

no comment

Additional comments

The authors have addressed all the comments I raised, I have no more comments.

Reviewer 4 ·

Basic reporting

I found the manuscript not well-written (major English language revision are necessary) and some aspects should be better detailed. Overall figure captions are very shorts, I recommend to detail them as much as possible. Figure quality should be improved as well.

Experimental design

I couldn’t perfectly understand how raw data were generated and processed

Validity of the findings

Nowadays, descriptions of mitochondrial genome assembly are not considered very interesting or fresh topics. They were fashionable long ago. However, I agree with the authors that the more molecular data available for certain groups, the more knowledge increases and I appreciated the effort to sequence the mtDNA of eleven different species.
Some further readings could better explain some parts of the discussion section

Annotated reviews are not available for download in order to protect the identity of reviewers who chose to remain anonymous.

Reviewer 5 ·

Basic reporting

• Strongly suggest to send the manuscript for English editing service.
• It would be great if a sentence or two on the background of leafhopper and the genus Batracomorphus is provided in the abstract
• Strongly suggest to state the taxonomic level for the taxa mentioned in the results, discussion and conclusion sections. For examples, the subfamily Iassinae, the tribe Macropsini, the family Cicadellidae and so on. These will ease the reader in understanding the writing.

Experimental design

More details need to be provided for methodology. Please see attached pdf.

Validity of the findings

More references to support the findings need to be provided. Please see attached pdf for more details.

Additional comments

• Suggestion on title: “The mitochondrial genome sequences of eleven leafhopper species of Batracomorphus (Hemiptera: Cicadellidae: Iassinae) unveil new gene rearrangements and phylogenetic implications”

Abstract
• It would be great if a sentence or two on the background of leafhopper and the genus Batracomorphus is provided in the abstract
• Line 13 – suggest correction “To determine the characteristics of the mitochondrial genomes and to elucidate the phylogenetic relationships between Batracomorphus species, we sequenced and compared the mitogenomes of 11 Batracomorphus species in this study.”
• Line 20 – the topologies of the six phylogenetic trees are essentially the same – please indicate what are the six trees involved
• Suggest keywords Iassinae and phylogenetic analysis instead of monophyletic and phylogeny

Methods
• Line 75 – Insect sampling and DNA extraction – the part on cox1 gene study is unclear. It is not known if a PCR of the cox1 gene was performed, and the gene sequence obtained was used for a BLAST search against NCBI database.
• Lines 79 to 80 – if voucher specimens were deposited in the institute, please provide the specimen voucher number for the 11 species. Also, if DNA samples were deposited, please describe the process for DNA sample preservation
• Lines 82 to 83 – as there is no mitochondrion enrichment prior DNA extraction, the authors should not mention that “the mitogenome of the 11 species were sequenced on the Illumina NovaSeq 6000…” Instead, it should be sequencing of the genomic DNA. Also, there should be a process to prepare the genomic DNA into library before sequencing in the NovaSeq. The reagent kit used and details of the process should be provided
• Line 85 – please indicate if there is quality-filtering of the raw sequence data before assembly of the mitogenomes
• Lines 87 to 90 – please indicate the rationale for comparing the assembled mitogenome sequences with T. arisana (GenBank: NC036480), a species from a different genus for homology.
• The MITOS link provided http://mitos.bioinf.uni-leipzig.de/index.py is no longer accessible. Please indicate if there is alternative webserver to access MITOS.
• The 12S and 16S rRNA genes should not be italicized and the S should be in capital. Please check throughout the manuscript
• Line 93 to 95 – MITOS webserver should be able to determine the rRNA genes from the mitogenomes. The method provided by the authors to determine their locations by the adjacent tRNA genes (trnV and trnL2) and via comparison with other Cicadellidae’s mitogenome sequences (not specified) in NCBI is vague.
• Lines 110 to 115 – please provide the rationale to use the first and second codon positions of the PCGs, and also the dataset coupled with the rRNA genes for phylogenetic analysis
• Please use 12S rRNA gene and 16S rRNA gene, instead of just 12S rRNA and 16S rRNA. Please check throughout the manuscript
• Line 119 – It is not clear what are the 4 datasets that were converted into ‘nex’ and ‘phy’ formats, as it was mentioned in previous section there were only 3 datasets, namely the first and second codon position for 13 PCGs, first and second codon position for 13 PCGs coupled with rRNA genes and all codon positions of 13 PCGs and rRNA genes
• Line 125 – please correct the sentence “The tree was stored every thousand generations”

Results
• Line 143 – the 4 species with gene rearrangements were not previously mentioned in the result section. Please describe this part of the results first before referring them as the 4 species with gene rearrangements
• Lines 149 to 152 – The A + T content of the 13 PCGs was between 77.9% (B. lineatus) and 80.1% (B. nigromarginattus), which showed a clear AT preference (the AT bias ranged from −0.13 to −0.09, with an average of −0.112) and a slight GC bias (the GC bias between −0.06 and −0.02, with an average of −0.037) (Fig. 2; Table S2). – this description is contradicting as the genes cannot be AT-biased and GC-biased at the same time
• Line 196 to 197 – A clear TA preference (the AT bias ranged from 0.07 to 0.12) and a slight CG bias (CG bias between −0.29 and 0.15) were observed– this description is contradicting as the genes cannot be AT-biased and GC-biased at the same time
• Line 217 – please provide reference to support the statement “tRNA gene rearrangements have also been reported in the mitochondrial genes of several other subfamilies of the leafhopper family”
• Lines 232 to 234 – The gene rearrangement in the mitogenomes of the Cicadellidae is relatively compact, with fewer and shorter overlapping regions and spacer regions between genes, but a few species were also observed having longer intergenic regions after gene rearrangements – to support this statement, please state which species
• Line 237 – please explain the statement “the number of spacers was the length was below 25 bp”
• Line 246 to 247 – please explain which parts of the results indicated the statement “The results of this study indicate that the phylogenetic relationship based on PCG12 data was more stable than that of the other two data sets. The reason for this difference may be the effect of the third codon of the PCGs in the mitogenomes on the phylogeny.”
• Line 250 – please include a description for the taxon Hylicinae in the statement “All phylogenetic trees showed that Eurymelinae formed paraphyletic groups.”
• Line 251 – the description Macropsini was stably clustered with Hylicinae was not observed in the phylogenetic tree
• Line 268 – Unfortunately, the status of B. fuscomaculatus and B. cornutus in the phylogenetic relationships remained questionable – please explain this statement

Discussion
• The discussion on the phylogenetic analysis repeated a lot of statements from the result sections
• Line 321 – Hu et al. (2022) analyzed the transcriptome data, showed that Iassinae was consistently polyphyletic (Hu et al. 2022). – the authors need to provide more explanation for this statement to tell how did the polyphyletic status of Iassinae was deduced from transcriptome data.
• Strongly suggest to state the taxonomic level for the taxa mentioned in the results, discussion and conclusion sections. For examples, the subfamily Iassinae, the tribe Macropsini, the family Cicadellidae and so on. These will ease the reader in understanding the writing.

Conclusion
• Please rewrite the sentence “a kind of gene rearrangement were founded in four species (B. allionii, B. extentus, B. lineatus, B. nigromarginatus), tRNA gene rearrangement events are hypothesized to include random gene loss after trnI-trnQ-trnM replication to form trnQ-trnI-trnM according to the TDRL model.”

Annotated reviews are not available for download in order to protect the identity of reviewers who chose to remain anonymous.

Reviewer 6 ·

Basic reporting

The manuscript needs considerable editing for language and writing style. The authors should have an English editing service to improve the manuscript. Need editing for typo errors, formatting of numbers and units (and the space between them) and grammar.

Experimental design

Introduction
Line 26 – 31, Line 36 - 44: Rearrange or rephrase the sentences to ensure the messages are conveyed more effectively.

Materials and Methods:
Line 63 – 64 “Firstly, 11 Batracomorphus species were identified by the male genitalia”: Please include citation.
Line 64 – 65: Describe molecular method to verify the accuracy of the morphological identification – the COI marker, PCR recipe and protocol used.
Line 72 – 74: Discuss the rationale of using different taxa for reference genome assembly and confirmation of sequence accuracy.

Validity of the findings

There is a lack of focus and significant detail on the genetic relationships of Batracomorphus, which should be one of the objectives of this study.
Line 271-272: What do you mean by “Macropsini and Idiocerini cannot be clustered together to present a sister group relationship”?
The title of this manuscript is “Mitochondrial genomes of Batracomorphus (Hemiptera: Cicadellidae: Iassinae) unveil new gene rearrangements with phylogenetic significance”, which, to me, should focus on gene arrangements and the phylogenetic significance in Batracomorphus. However, under the section "Implications of phylogeny", the discussion primarily revolves around the family Cicadellidae.
The authors also mentioned in the Introduction that the research gap lies in the: (i) similarities of morphological characteristics within the genus Batracomorphus; and (ii) the use of traditional taxonomy for systematic studies of this genus. However, I did not see these aspects being thoroughly discussed. Which species of Batracomorphus exhibit similarities based on morphological features that led to this molecular study?

---

## Round 0.3 · Minor Revisions

Thank you for your submission to PeerJ. Please change the manuscript as per the comments of the reviewer.

Reviewer 5 ·

Basic reporting

The manuscript has been greatly improved but there are still some minor mistakes that need to be taken care of.

Experimental design

The correction done is satisfactory but still there are some minor mistakes that require the attention of the authors. Please see additional comments. The manuscript can be accepted provided the minor mistakes are attended to.

Validity of the findings

More references have been provided and the discussion is much improved now.

Additional comments

Abstract
• It would be great if a sentence or two on the background of leafhopper and the genus Batracomorphus is provided in the abstract
• Line 19 – The topologies of six phylogenetic trees were in agreement. – please indicate what are the six trees involved, otherwise, just tell that all phylogenetic analyses

Introduction
• Line 40 – Please correct “Recent, studies”

Methods
• Lines 79 to 80 – It is not clear if insect specimens or their DNA were deposited in the institute. If voucher specimens were deposited in the institute, please provide the specimen voucher number for the 11 species. Also, if DNA samples were deposited, please describe the process for DNA sample preservation
• Line 85 – please indicate if there is quality-filtering of the raw sequence data before assembly of the mitogenomes
• Lines 87 to 90 – please indicate the rationale for comparing the assembled mitogenome sequences with T. arisana (GenBank: NC036480), a species from a different genus for homology to confirm the accuracy of the sequence
• The MITOS link provided http://mitos.bioinf.uni-leipzig.de/index.py is no longer accessible. Please indicate if there is alternative webserver to access MITOS.
• Line 125 – please explain the meaning of the sentence “The tree was stored every thousand generations”

Results
• Lines 148 to 151 – The A + T content of the 13 PCGs was between 77.9% (B. lineatus) and 80.1% (B. nigromarginattus), which showed a clear AT preference (the AT bias ranged from −0.13 to −0.09, with an average of −0.112) and a slight GC bias (the GC bias between −0.06 and −0.02, with an average of −0.037) (Fig. 2; Table S2). – this description is contradicting as the genes cannot be AT-biased and GC-biased at the same time
• Line 189 – please correct aftertrnI
• Line 194 to 196 – A clear TA preference (the AT bias ranged from 0.07 to 0.12) and a slight CG bias (CG bias between −0.29 and 0.15) were observed– this description is contradicting as the genes cannot be AT-biased and GC-biased at the same time
• Line 213 – “Finally, a new gene sequence trnQ-trnI-trnM was generated” should be a new gene order, instead of gene sequence
• Line 246 – please correct the 3rd in superscript
• Line 250 – please include a description for the taxon Hylicinae in the statement “All phylogenetic trees showed that Eurymelinae formed paraphyletic groups.”
• Line 251 – the description Macropsini was stably clustered with Hylicinae was not observed in the phylogenetic tree
• Line 305 – please correct “a well-supported monophyl etic status”
• Line 316 – 28S should not be italicized

Figure 4 – please correct Duplicatedregion

---

## Round 0.4 · accepted · Accept

Thanks for your work to PeerJ.
Congratulations
Sorry it took long for the review proceedings.
The result is better.
Sorry again.